# The Effect Sizes of *PPARγ* rs1801282, *FTO* rs9939609, and *MC4R* rs2229616 Variants on Type 2 Diabetes Mellitus Risk among the Western Saudi Population: A Cross-Sectional Prospective Study

**DOI:** 10.3390/genes11010098

**Published:** 2020-01-14

**Authors:** Sherin Bakhashab, Najlaa Filimban, Rana M. Altall, Rami Nassir, Safaa Y. Qusti, Mohammed H. Alqahtani, Adel M. Abuzenadah, Ashraf Dallol

**Affiliations:** 1Department of Biochemistry, Faculty of Science, King Abdulaziz University, P.O Box 80218, Jeddah 21589, Saudi Arabia; n3519339@kfshrc.edu.sa (N.F.); om_hanad@hotmail.com (R.M.A.); squsti@kau.edu.sa (S.Y.Q.); 2Center of Innovation in Personalized Medicine, King Abdulaziz University, P.O Box 80216, Jeddah 21589, Saudi Arabia; aabuzenadah@kau.edu.sa (A.M.A.); adallol@kau.edu.sa (A.D.); 3King Faisal Specialist Hospital and Research Center, Clinical Genomics, Department of Genetics, P.O. Box 3354, Riyadh 11211, Saudi Arabia; 4Department of Pathology, Faculty of Medicine, Umm Al-Qura University, P.O. Box 715, Makkah 21955, Saudi Arabia; rmnassir@ucdavis.edu; 5Department of Medical Laboratory Technology, Faculty of Applied Medical Sciences, King Abdulaziz University, P.O Box 80216, Jeddah 21589, Saudi Arabia; mhalqahtani@kau.edu.sa; 6Center of Excellence in Genomic Medicine Research, King Abdulaziz University, P.O Box 80216, Jeddah 21589, Saudi Arabia

**Keywords:** type 2 diabetes mellitus, single nucleotide polymorphism, *PPARγ*, *FTO*, *MC4R*, obesity

## Abstract

Type 2 diabetes mellitus (T2DM) is a common polygenic disease with associated comorbidities. Obesity is a major risk factor for the development of T2DM. The aim of this study is to determine the allele and genotype frequency of peroxisome proliferator-activated receptor-*γ* (*PPARγ*) rs1801282, fat mass and obesity-associated protein (*FTO*) rs9939609, and melanocortin 4 receptor (*MC4R*) rs2229616 polymorphisms and their association with risk of T2DM in the western Saudi population as mediators of adiposity phenotypes. In a cross-sectional prospective study, genomic DNA from control and T2DM patients were isolated and genotyped for these single-nucleotide polymorphisms. There was a significant association of the *MC4R* rs2229616 variant with T2DM, but no association with T2DM was detected with *PPARγ* rs1801282 or *FTO* rs9939609. The combination of C/C for *PPARγ* rs1801282, A/A for *FTO* rs9939609, and C/C for *MC4R* rs2229616 increased the risk of T2DM by 1.82. The A/T genotype for *FTO* rs9939609 was predicted to decrease the risk of T2DM when combined with C/C for *PPARγ* rs1801282 and C/C for *MC4R* rs2229616 or C/C for *PPARγ* rs1801282 and C/T *MC4R* rs2229616. In conclusion, our study showed the risk of the assessed variants for the development of T2DM in the Saudi population.

## 1. Introduction

Type 2 diabetes mellitus (T2DM) is a major health burden in Saudi Arabia. A recent national epidemiological health survey revealed that 23.7% of Saudis living in developed areas have T2DM [1,2,3]. T2DM is a complex disorder involving interactions between genetic and environmental factors, with obesity being a primary risk factor [4]. Body mass index (BMI) is considered the most important phenotypic trait in studying individual genetic predisposition for obesity because it measures fat-free mass and fat mass [5]. Various studies have established that genetic susceptibility of T2DM is polygenic, and genome-wide association studies (GWAS) have detected more than 20 loci associated with the risk of T2DM [6,7].

Rapid development in genotyping technology has led to the discovery of further genetic variants responsible for the increased risk of T2DM. In a study by Majithia et al., 49 variants with decreased function in the peroxisome proliferator-activated receptor-*γ* (*PPARγ)* gene were identified by large-scale sequencing of 19,752 participants with or without T2DM, 1 in 1000 cases was at risk to develop T2DM [8]. *PPARγ,* a transcription factor, belongs to the nuclear hormone receptor family mediating adipocyte differentiation and glucose homeostasis. It has two isoforms: *PPARγ*1, expressed across different tissues, and *PPARγ*2, highly expressed in adipocytes [9]. The *PPARγ* gene is located on chromosome 3 and has many variants [10]. In a meta-analysis, the common C allele (85%) was reported to moderately (odds ratio, OR = 1.25) increase susceptibility to T2DM [11]. It was estimated that 75% of individuals of Caucasian origin are homozygous CC (Pro12 allele) for T2DM [11]. In an attempt to establish T2DM-risk polymorphisms using previously known loci, *PPARγ2* rs1801282 C>G (Pro12Ala) was found to be the most common genetic variant, resulting in a proline to alanine substitution at amino acid residue position 12 [12]. As *PPARγ* is involved in adipogenesis, reduction in the transcription of *PPARγ* decreases lipid uptake and improves insulin resistance and inhibition of glucose production [10].

GWAS using the genome-wide human single nucleotide polymorphism (SNP) array, containing 440794 SNPs, to predict the early onset of obesity among young Germans highlighted the substantial contribution of variations in fat mass and obesity-associated protein (*FTO*) in the initial progression to obesity [13]. In a population-based cohort study of white Europeans, *FTO* rs9939609 was found to be the SNP most associated with an increase in BMI in patients with T2DM. This SNP is located within 47 kb of the first intron of the *FTO* gene. In addition, individuals who were homozygous for the A allele of *FTO* rs9939609 (16% of the population) were found to be at high risk for gaining weight or being obese. In contrast, individuals homozygous for the T allele (37% of the population) were at low risk for T2DM [14]. The *FTO* gene is located at chromosome 16q12.2 and contains variations in its intronic regions that may be associated with an increase in body fat, leading to a potential risk of obesity [15]. It has been suggested that it regulates the appetite and body weight, as it is highly expressed in the hypothalamus. In addition, it may be involved in nucleic acid demethylation, mediating the transcription of genes responsible for energy metabolism [16,17]. These findings were followed by a series of studies that genotyped *FTO* rs9939609 in several populations with different ethnic backgrounds [18,19,20,21]. Previous association studies on Arab populations about the contribution of *FTO* rs9939609 to the risk of T2DM, one in the middle Saudi population [22] and two among the Iraqi population [23,24], showed divergent results. The mechanistic basis for the genetic association between *FTO* and obesity appears to be due to the disruption of AT-Rich Interaction Domain 5B repressor through the causal variants of *FTO*. The disruption results in loss of binding and activation of downstream targets Iroquois Homeobox 3 (*IRX3*) and *IRX5* during early adipocyte differentiation [25].

The melanocortin 4 receptor (*MC4R*) gene, located at chromosome 18q21, encodes a G protein-coupled receptor that binds to the α-melanocyte-stimulating hormone. It plays a role in fatty acid metabolism in the liver (promoting lipolysis) and lowering insulin secretion in the pancreas [26]. *MC4R* is expressed in the hypothalamus in response to the leptin signaling pathways [27,28]. It is believed that *MC4R* induces lipolysis directly in rabbit adipocytes [29] and at a minimum level in subcutaneous adipocytes of obese humans [30], mediated by the intracellular response to neuronal signals [26]. Association signals were identified between *MC4R* exonic variants and different levels of obesity. Rare mutations in the *MC4R* gene, in which adiposity presents a form of the monogenic hereditary disorder, have been detected [31,32,33]. Several population studies have generated similar evidence demonstrating the association of *MC4R* rs2229616 Val103Ile with protection against obesity and increasing MC4R lipolysis activity and eventual reduction in body weight [34]. Conversely, a non-coding region located at 3′ of MC4R might harbor variants related to obesity risk, according to a GWAS analysis of the European population [35].

Lifestyle changes occurring as a result of post-revolution social transformation may be responsible for the increase in the percentages of the Saudi population affected with obesity, T2DM, and metabolic syndrome [36]. Thus, we aimed in this population study to determine the effect of *PPARγ* rs1801282, *FTO* rs9939609, and *MC4R* rs2229616 variants on the risk of T2DM among the western Saudi population as mediators of adiposity phenotypes. The risk of these variants has not been reported previously among Saudis in the western region.

## 2. Materials and Methods

### 2.1. Study Design

A cross-sectional prospective study was conducted in King Abdulaziz University Hospital and Jeddah Care Center for Diabetes and Hypertension, Jeddah, Saudi Arabia from 2016 to 2018. The total sample size was 738 cases after applying the exclusion criteria. Patients with other chronic diseases, including cardiovascular disease and metabolic syndrome, were excluded. The power of the statistic was 95% calculated for a sample size of at least 571 and an OR = 1.2 for the studied SNPs using G*Power software. In this study, data from 415 diabetic cases and 323 controls were included. All participants in the current study, both cases and controls were identified as Saudi based on self-reported ethnicity, which should present itself as a homogenous sample set genetically. Having a population with such a homogenous ancestry would decrease the type 2 error rate [37]. The study was approved by the Biomedical Ethics Unit, Faculty of Medicine, KAU (Approval Number 371-16), and the Ministry of Health (Approval Number A00363). Written informed consent was obtained from participants prior to sample collection. The study was conducted in accordance with the Declaration of Helsinki.

Biochemical tests, HbA1c, fasting blood sugar (FBS), triglyceride, and cholesterol were assessed. The T2DM patients were diagnosed according to the American Diabetes Association criteria and were under treatment with metformin and/or other T2DM drugs. Clinical parameters, such as age, body weight, height, and BMI were recorded.

### 2.2. Genotyping

Genomic DNA was isolated from peripheral whole blood using the QIAamp Blood Maxi Kit (QIAGEN, Hilden, Germany) following the manufacturer’s protocol. All samples were genotyped for *PPARγ* rs1801282 (assay ID: C_1129864_10), *FTO* rs9939609 (assay ID: C_30090620_10), and *MC4R* rs2229616 (assay ID: C_15851270_20) using TaqMan SNP Genotyping Assays (Thermo Fisher Scientific, Waltham, MA, USA). Allelic discrimination plots were analyzed using the QuantStudio 12K Flex Real-Time PCR System (Thermo Fisher Scientific).

### 2.3. Statistical Analysis

Quantitative data are presented as (mean ± SD), whereas qualitative data are presented as frequencies. Genotypes and allelic frequencies were calculated within the cohort population to determine the significant differences between the T2DM patient and control groups. Clinical variables (BMI, FBS, HbA1c, triglycerides, and cholesterol) were further classified into categories (Table 1). Pearson’s chi-squared test was used to test the association of the studied variants with T2DM, considering *p* < 0.05 as being statistically significant. Further, the relative risk for alleles was calculated to determine the risk allele in each variant. A binary logistic regression analysis was performed to study the association between variables and the study groups. The impact of the combined SNP genotypes on the risk of T2DM was determined using binary logistic regression by apportioning effect size between dependent and independent factors. Data analysis was conducted using IBM SPSS Statistics software, Version 24 (IBM Corp., Armonk, NY, USA).

## 3. Results

### 3.1. The Association of PPARγ rs1801282, FTO rs9939609, and MC4R rs2229616 Variants with T2DM Risk

Clinical features of control and case subjects are listed in Table 2. Minor allele frequencies for *PPARγ* rs1801282, *FTO* rs9939609, and *MC4R* rs2229616 are presented in Table 3. The data showed a considerable difference in *PPARγ rs1801282* between the rare G allele (0.08 control and 0.07 T2DM) and C allele (0.92 control and 0.93 T2DM). With regard to *FTO* rs9939609, no differences in the A allele (0.39 control and 0.46 T2DM group) and T allele (0.61 control and 0.54 T2DM) frequencies were detected. However, a higher frequency (0.99) of the C allele in the T2DM group was observed than in the control group (0.88) for *MC4R* rs2229616. The T allele frequency was 0.12 for the control and 0.01 for the T2DM group. To explore how the genetic components affect the risk of diabetes, we assessed the association of each variant with T2DM by Pearson’s chi-squared test. A significant association of *MC4R* rs2229616 was observed and was involved in individual susceptibility to T2DM (*p* < 0.001). However, no significant association of *PPARγ* rs180128 and *FTO* rs9939609 with the risk of T2DM was detected (Table 3).

### 3.2. Relative Risk for T2DM from Alleles in PPARγ rs1801282, FTO rs9939609, and MC4R rs2229616 SNPs 

To confirm the risk allele for the studied variants, the relative risk was calculated, showing C as the risk allele in *PPARγ* rs1801282. It was correlated with T2DM 1.16 times more than in the control group (Table 4). In *FTO* rs9939609, A was the risk allele, correlated to T2DM 1.07 more times than the control group. The C allele in *MC4R* rs2229616 was highly correlated to T2DM, 6.56-fold more than the control.

### 3.3. Associations of Categorical Factors with the Risk of T2DM in PPARγ rs1801282, FTO rs9939609, and MC4R rs2229616 SNP Genotypes

To explore how the genetic components affected the risk of T2DM, we assessed the influence of each SNP genotype for *PPARγ* rs1801282, *FTO* rs9939609, and *MC4R* rs2229616. No significant correlation for all SNPs genotypes was detected with T2DM. The effect of all three SNPs: *PPARγ* rs1801282, *FTO* rs9939609, and *MC4R* rs2229616 loci on the categorical clinical parameters was examined by binary logistic regression (Table 5). There was no significant effect of studied clinical parameters (obesity, HbA1c, FBS, triglycerides, and cholesterol) on the risk of T2DM in all SNPs genotypes.

### 3.4. The Risk for T2DM from Combined Polygenic SNPs, PPARγ rs1801282, FTO rs9939609, or MC4R rs2229616

The effect of combined genotypes of the three SNPs, *PPARγ* rs1801282, *FTO* rs9939609, and *MC4R* rs2229616, on the risk of T2DM was determined by binary logistic regression. The existence of C/C for *PPARγ* rs1801282, A/A for *FTO* rs9939609, and C/C for *MC4R* rs2229616 together significantly increased the risk of T2DM by 1.82, *p* = 0.045. The A/T genotype for *FTO* rs9939609 variant was more likely to be protective from the risk of T2DM when combined with either C/C for *PPARγ* rs1801282 and C/C for *MC4R* rs2229616 (OR = 0.25, *p* = 0.097) non-significantly or C/C for *PPARγ* rs1801282 and C/T *MC4R* rs2229616 (OR =0.04, *p* = 0.002, Table 6) significantly. No change on the risk of T2DM was observed with other combined genotypes.

## 4. Discussion

In this genetic association study, we evaluated whether the *PPARγ* rs1801282, *FTO* rs9939609, and *MC4R* rs2229616 SNPs were associated with the risk of T2DM among the western Saudi population, and their association with the clinical factors that give rise to T2DM. In addition, we studied the effect of combined genotypes on the risk of T2DM. These variants were selected because of their importance as mediators of adiposity. To our knowledge, this is the first study reporting the effect sizes of these variants on T2DM risk in the western region of Saudi Arabia. All participants in this study were Saudi, as this was more likely to be a homogeneous sample genetically.

Over the past decade, genome-wide associations have rapidly become a preferred approach for investigators who have identified thousands of SNPs [12]. The significance of the associations depends greatly on the success of the SNP replication study in different ethnic groups. Considering SNPs as predictors for the disease, several attempts were required to confirm the association signals [38]. In our study, the frequency of the common C allele of *PPARγ* rs1801282 (Pro12Ala) was 0.92 for the control and 0.93 for the T2DM group. The G minor allele frequency for controls and T2DM group was respectively 0.08 and 0.07, a similar frequency (0.07) to that reported by the 1000 genome project. There was no association detected between this SNP and T2DM. In concordance with our study, *PPARγ* rs1801282 was not associated with the risk of T2DM in the Emirati population [39]. The frequencies of the *PPARγ* Pro12Ala genotypes were 91.9% for C/C, 8.1% for C/G and 0% for G/G in the Emirati population [39], compared to 87.3%, 11.6%, and 1.1%, respectively, in the diabetic group data in the present study. Similar to our findings here, recent studies in Iraqi and Egyptian populations reported no significant association in their cohorts with T2DM [40,41]. In addition, the Egyptian study showed no significant differences in *PPARγ* Pro12Ala genotypes and T2DM parameters (BMI, HbA1c, FBS, triglyceride, and cholesterol) [40]. 

Unlike *PPARγ* rs1801282, *FTO* rs9939609 is an intronic variant that was first identified via GWAS to be associated with obesity, conferring risk for T2DM [42]. In an attempt to understand the mechanism of *FTO* obesity modulation, it was suggested that an individual’s behavior toward food consumption is primarily regulated by this SNP. In fact, it suggests that A/A homozygous is more likely to be responsible for the reduction of personal ability to perform regular activity [43]. The A allele was reported as the risk allele [12], which is consistent with the present study. Variations in frequency are driven by ethnic differences and environmental stimuli. The minor A allele frequency was relatively higher in controls (0.39) and diabetic groups (0.46) than what was reported in the 1000-genome project (0.34) but was consistent with the European adults who are diabetic or obese (0.45) [14]. In a longitudinal study conducted to examine the association of various clinical parameters with the incidence of cardiovascular diseases and T2DM, the A allele frequency for *FTO* rs9939609 in White, Black, Hispanic, and Asian/Pacific Islanders for the case/control group was 42.3/39.7, 47.1/44.7, 27.0/30.7, 20.1/16.9, respectively, with no involvement in T2DM risk [44]. The frequency of this homozygous form (A/A) risk allele was slightly higher in the current study (0.2) than that measured in the European population (0.16) [14]. There was no significant association between *FTO* SNP and the risk of diabetes as an independent variable. Another study on the Saudi population in the middle region was conducted on the same SNP showed no correlation with T2DM, similar to the current study [22]. However, contrary to our results, two Iraqi studies revealed a significant association of *FTO* rs9939609 variant with the risk of T2DM in obese patients [23,24]. The overall impact of *FTO* SNP on adiposity and T2DM is still elusive because of a lack of knowledge on its individual or combined role with other genes harboring the same locus [14,19,45,46].

The third studied gene, *MC4R*, is mainly expressed in the brain and spinal cord and has a role in the regulation of metabolic and dieting behavior and energy homeostasis [29]. A recent genomic association study conducted within the European population revealed 61 non-synonymous variants in *MC4R*, of which 77% produce a decreased level of MC4R that is insufficient to bind to β-arrestin-2 and increases BMI and risks of obesity and T2DM [47]. *MC4R* rs2229616 C>T is a missense variant Val103Ile located in the coding region of chromosome 18 that provides vital protection against severe obesity [48,49]. One functional study found that the 103Ile allele (T allele) reduces the endogenous MC4R antagonist, agouti-related protein (AGRP), and increases MC4R agonist effectiveness [34]. Thus, the *MC4R* rs2229616 variant exhibits protective effects resulting in lower body weight [34,50]. In a meta-analysis conducted on the Caucasian population, a remarkable reduction of BMI by 0.52 was found in individuals who are carriers of heterozygote *MC4R* rs2229616 C/T [51]. Another epidemiological study showed a strong association of this *MC4R* variant with decreasing the waist circumference and HbA1c level and increasing HDL-cholesterol [52]. This research elucidated the *MC4R* Val103Ile association with obesity-related phenotypes (BMI and HbA1c levels) [52]. Compared to global minor allele frequency for the T allele (0.016), the frequency of this allele in our study population was quite low (0.01) in the T2DM group and higher in the control group (0.13). Since the T allele might have a protective effect, individuals who lack this allele are more susceptible to higher BMI and development of diabetes due to uncontrollable appetite and physical inactivity. Clearly, the frequency of the C allele is higher in the T2DM group (0.99) than in the control group (0.88). We found a strong association of *MC4R* rs2229616 variant with T2DM risk similar to the results from a study of patients of Chinese Han ancestry with T2DM [53]. No study on this *MC4R* SNP had been conducted on T2DM patients in the Arab populations.

Additionally, we assessed the mutual effect of polygenic risk of the three variants’ genotypes to show that the C/C for *PPARγ* rs1801282, A/A for *FTO* rs9939609, and C/C for *MC4R* rs2229616 increased the risk susceptibility of T2DM. Whereas the A/T genotype for *FTO* rs9939609 was predicted to be a modifier that can overcome the risk of C/C for *PPARγ* rs1801282 and C/C or C/T for *MC4R* rs2229616 when the three genotypes combined. Homozygosity or heterozygosity of T allele for *FTO* rs9939609 had a protective effect, lowering the risk of T2DM when inherited collectively with *PPARγ* rs1801282 and *MC4R* rs2229616 variants.

## 5. Conclusions

Taken all together, these SNPs were associated with T2DM mainly if they were inherited mutually with genotypes C/C for *PPARγ* rs1801282, A/A for *FTO* rs9939609, and C/C for *MC4R* rs2229616 in the western Saudi population. The T allele for *FTO* rs9939609 was revealed to have a negative effect on the development of T2DM when combined with C/C *PPARγ* rs1801282 and C/C or C/T *MC4R* rs2229616 genotypes. In addition, the *MC4R* rs2229616 variant independently contributed to the risk of T2DM among Saudis in the western region.

## Figures and Tables

**Table 1 genes-11-00098-t001:** Classification of variables into different categories.

Variable	Category
Obesity (BMI) (kg/m^2^)	Non-Obese (<25)
	Obese (≥25)
FBS (mmol/l)	Controllable (<7)
	Uncontrollable (≥7)
HbA1c (%)	Controllable (<7)
	Uncontrollable (≥7)
Triglyceride level (mmol/l)	Desirable (<2.2)
	High risk (≥2.2)
Cholesterol level (mmol/l)	Desirable (<6.2)
	High risk (≥6.2)

BMI: Body mass index, FBS: Fasting blood glucose, HbA1c: Glycated hemoglobin test.

**Table 2 genes-11-00098-t002:** Clinical characteristics of type 2 diabetic and control subjects.

Variable	Control Sample (n = 323)	Diabetic Sample (n = 415)	*p*-Value
Age (years)	34.4 ± 15.5	56.2 ± 12.9	<0.001
Height (cm)	162.5± 9.8	162.5 ± 9.1	0.970
Weight (kg)	67.9 ± 18.5	77.6 ± 15.2	<0.001
BMI (kg/m^2^)	25.3 ± 6.1	30.1 ± 7.0	<0.001
FBS (mmol/l)	5.9 ± 2.1	8.2 ± 3.6	<0.001
HbA1c (%)	6.0 ± 1.1	8.2 ± 2.0	<0.001
Triglycerides (mmol/l)	0.74 ± 0.49	1.1 ± 0.75	<0.001
Cholesterol (mmol/l)	4.73 ± 1.14	4.66 ± 1.16	0.654

Values are expressed as mean ±SD. *p*-values were calculated by Student’s *t*-test. A *p*-value < 0.05 was considered statistically significant. BMI: Body mass index, FBS: Fasting blood glucose, HbA1c: Glycated hemoglobin test.

**Table 3 genes-11-00098-t003:** Association signals identified for each single nucleotide polymorphism (SNP) for type 2 diabetes mellitus (T2DM) risk.

SNP	Genotype	Genotype Frequency (CTRL)	Genotype Frequency (T2DM)	*p*-Value	Minor Allele	1000-Genome MAF	CTRL MAF	T2DM MAF
*PPARγ* rs1801282 Pro12Ala	C/C	225 (83.6%)	308 (87.3%)	0.348	G	0.07	0.08	0.07
	C/G	38 (14.1%)	41 (11.6%)					
	G/G	6 (2.2%)	4 (1.1%)					
*FTO* rs9939609 intronic variant	A/A	51 (16.2%)	73 (20.0%)	0.367	A	0.34	0.39	0.46
	A/T	168 (53.3%)	194 (53.0%)					
	T/T	96 (30.5%)	99 (27%)					
*MC4R* rs2229616 Val103Ile	C/C	170 (75.9%)	337 (98.3%)	<0.001	T	0.016	0.12	0.01
	C/T	52 (23.2%)	6 (1.7%)					
	T/T	2 (0.9%)	0 (0%)					

*p*-values were calculated by Pearson’s chi-squared test. *p*-values < 0.05 were considered statistically significant. CTRL: Controls, T2DM: Type 2 diabetes mellitus, MAF: Minor allele frequency.

**Table 4 genes-11-00098-t004:** Relative risk for T2DM from alleles in *PPARγ* rs1801282, *FTO* rs9939609, and *MC4R* rs2229616 compared to control.

SNP	Allele	Allele Frequency (T2DM)	Allele Frequency (CTRL)	RR	OR	*p*-Value
*PPARγ rs1801282*	C G	657 49	488 50	1.16	1.37	0.139
*FTO rs9939609*	A T	340 392	270 360	1.07	1.16	0.190
*MC4R rs2229616*	C T	680 6	392 56	6.56	16.19	<0.0001

*p*-values were calculated by chi-square test (2 x 2 contingency chi-square test) and corrected by the Yates correction. *p*-values < 0.05 were considered statistically significant. OR: Odds ratio, RR: Relative risk.

**Table 5 genes-11-00098-t005:** Associations between each categorical factor and the risk of T2DM for each SNP genotype on T2DM compared to the control group.

SNP	Factors	Genotype	*p*-Value	OR	95% C.I. for OR	
					Lower	Upper
*PPARγ* rs1801282 Pro12Ala	T2DM	C/C	0.269	2.05	0.57	7.36
		C/G	0.481	1.62	0.42	6.18
	Obesity (BMI) ≥ 25	C/C	0.592	1.72	0.24	12.38
		C/G	0.806	1.29	0.17	10.15
	HbA1c (%)	C/C	0.999	0.0	-	-
		C/G	0.999	0.0	-	-
	FBS (mmol/l)	C/C	1.000	0.0	0.0	-
		C/G	1.000	0.0	0.0	-
	Triglyceride level (mmol/l)	C/C	1.000	0.0	-	-
	Cholesterol level (mmol/l)	C/C	-	-	-	-
*FTO* rs9939609 intronic variant	T2DM	A/A	0.158	1.39	0.88	2.19
		A/T	0.525	1.12	0.79	1.59
	Obesity (BMI) ≥ 25	A/A	0.882	0.95	0.49	1.85
		A/T	0.580	0.86	0.51	1.46
	HbA1c (%)	A/A	0.431	1.97	0.37	10.59
		A/T	0.618	1.34	0.42	4.27
	FBS (mmol/l)	A/A	0.697	1.36	0.29	6.32
		A/T	0.276	2.18	0.54	8.80
	Triglyceride level (mmol/l)	A/A	0.999	-	0.0	-
		A/T	0.826	1.40	0.07	28.12
	Cholesterol level (mmol/l)	A/A	0.427	0.33	0.02	5.03
		A/T	0.196	3.67	0.51	26.22
*MC4R* rs2229616 Val103Ile	T2DM	C/C	0.999	-	-	-
		C/T	0.999	-	-	-
	Obesity (BMI) ≥ 25	C/C	1.000	-	-	-
		C/T	1.000	-	-	-
	HbA1c (%)	C/C	0.999	0.0	0.0	-
	FBS (mmol/l)	C/C	1.000	0.0	0.0	-
	Triglyceride level (mmol/l)	C/C	-	-	-	-
	Cholesterol level (mmol/l)	C/C	-	-	-	-

*p*-values were calculated by binary logistic regression. G/G, T/T and T/T are the reference genotypes for *PPARγ* rs1801282, *FTO* rs9939609, and *MC4R* rs2229616, respectively. BMI < 25 kg/m^2^, FBS < 7 mmol/l, HbA1c < 7%, triglyceride < 2.2 mmol/l, and cholesterol < 6.2 mmol/l are the reference categorical factors. C.I.: Coefficient interval, OR: Odds ratio.

**Table 6 genes-11-00098-t006:** The impact of combined SNPs genotypes on the risk of T2DM.

Combined SNPs Genotypes	*p*-Value	OR	95% C.I. for OR	
			Lower	Upper
*PPARγ* rs1801282 (C/C) by *FTO* rs9939609 (A/A) by *MC4R* rs2229616 (C/C)	0.045 ^*^	1.82	1.01	3.27
*PPARγ* rs1801282 (C/C) by *FTO* rs9939609 (A/A) by *MC4R* rs2229616 (C/T)	0.084	1.45	0.95	2.20
*PPARγ* rs1801282 (C/C) by *FTO* rs9939609 (A/T) by *MC4R* rs2229616 (C/C)	0.097	0.25	0.05	1.29
*PPARγ* rs1801282 (C/C) by *FTO* rs9939609 (A/T) by *MC4R* rs2229616 (C/T)	0.002 ^**^	0.04	0.01	0.30
*PPARγ* rs1801282 (C/T) by *FTO* rs9939609 (A/A) by *MC4R* rs2229616 (C/C)	0.827	0.88	0.28	2.74
*PPARγ* rs1801282 (C/T) by *FTO* rs9939609 (A/A) by *MC4R* rs2229616 (C/T)	0.829	0.91	0.37	2.22
*PPARγ* rs1801282 (C/T) by *FTO* rs9939609 (A/T) by *MC4R* rs2229616 (C/T)	1.000	0.00	0.000	-

*p*-values were calculated by binary logistic regression. G/G, T/T and T/T are the reference genotypes for *PPARγ* rs1801282, *FTO* rs9939609, and *MC4R* rs2229616, respectively. ^*^
*p* < 0.05, ^**^
*p* < 0.01.

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
