# Peer review of "The Effect Sizes of PPARγ rs1801282, FTO rs9939609, and MC4R rs2229616 Variants on Type 2 Diabetes Mellitus Risk among the Western Saudi Population: A Cross-Sectional Prospective Study"

_genes, 2020, doi:10.3390/genes11010098_

Round 1
Reviewer 1 Report
I thank the authors for responding to my comments. There are two major comments that have not been addressed:
In my "Point 4", there were actually two requests:
The authors should control for potential population structure in their regression model, by incorporating either genomic principal components or genetic relationship matrices (GRM)
The authors should if possible control for potential population structure using one of the commonly used methods for doing so. E.g. PMID 20548291
p<0.05 is not the correct statistical threshold given that multiple SNPs and multiple traits are tested. A Bonferroni significance threshold should be used
Since the authors perform more than one test, they need to use a significance threshold of (at least) 0.05 / (# of tests)
2. In addition, the citations are still not fully correct:
The initial report of the PPARG association as Altshuler et al 2000 The initial report of FTO was Frayling et al 2007 The initial report of MC4R was Loos et al 2008 The authors should also cite Claussnitzer et al 2015 regarding FTO since in that reference they argue that IRX3/5, rather than FTO, is the causal gene for the FTO signal The teference should be "Majithia et al", not "Amit et al"Author Response
Response to Reviewer 1 Comments
Comments and Suggestions for Authors
I thank the authors for responding to my comments. There are two major comments that have not been addressed:
Point 1: In my "Point 4", there were actually two requests:
The authors should control for potential population structure in their regression model, by incorporating either genomic principal components or genetic relationship matrices (GRM). The authors should if possible control for potential population structure using one of the commonly used methods for doing so. E.g. PMID 20548291. p<0.05 is not the correct statistical threshold given that multiple SNPs and multiple traits are tested. A Bonferroni significance threshold should be used Since the authors perform more than one test, they need to use a significance threshold of (at least) 0.05 / (# of tests).
Response 1: Thank you for this comment. This point was justified in lines 126-129 in the manuscript. We added statistical geneticist to the co-authorship who revised the statistics in the manuscript, the analyses were changed in Tables 3, 4, 5 and 6.
Point 2: In addition, the citations are still not fully correct:
The initial report of the PPARG association as Altshuler et al 2000 The initial report of FTO was Frayling et al 2007 The initial report of MC4R was Loos et al 2008 The authors should also cite Claussnitzer et al 2015 regarding FTO since in that reference they argue that IRX3/5, rather than FTO, is the causal gene for the FTO signal The teference should be "Majithia et al", not "Amit et al"
Response 2: Thank you for this comment. The initial report of the PPARG association as Altshuler et al 2000 was in the introduction (lines 69-70) and it is the initial report as stated. The initial report of FTO was Frayling et al 2007 was moved to lines 79-85 (highlighted in yellow). The initial report of MC4R was Loos et al 2008 was added (reference 33). We had cited Claussnitzer et al 2015, highlighted in yellow reference 25. The reference "Majithia et al" had been corrected.
Research design had been improved. The introduction, results, discussion and conclusion had been amended according to the new analysis. The manuscript had been revised by English native speaker editor from Cambridge proofreading and editing LLC.
Reviewer 2 Report
I sincerely believe the authors need a statistical geneticist to have a look at the paper and re-do some of the analyses in light of our previous comments. They have not addressed any of my - or the other reviewers' - comments as it seems like they are not aware of the statistical methods used by the top papers published in the field. I think the authors should look into the latest T2D GWASs and try and emulate the analyses done there.
Author Response
Comments and Suggestions for Authors
I sincerely believe the authors need a statistical geneticist to have a look at the paper and re-do some of the analyses in light of our previous comments. They have not addressed any of my - or the other reviewers' - comments as it seems like they are not aware of the statistical methods used by the top papers published in the field. I think the authors should look into the latest T2D GWASs and try and emulate the analyses done there.
Response: Thank you for your comment. We added Dr. Rami Nassir as co-author, a statistical geneticist who worked in the University of California, Davis. He had many publications on population and genome-wide association studies. Dr. Nassir revised the manuscript and redo some of the analyses as suggested. The analyses were changed in Tables 3, 4, 5 and 6.
Research design had been improved. The introduction, results, discussion and conclusion had been amended according to the different results. The manuscript had been revised by English native speaker editor from Cambridge proofreading and editing LLC.
Reviewer 3 Report
The authors answered the questions.
Author Response
Comments and Suggestions for Authors
The authors answered the questions.
Response: Thank you, no comments were addressed by this reviewer.
Research design had been improved. The introduction, results, discussion and conclusion had been amended according to the different results. The manuscript had been revised by English native speaker editor from Cambridge proofreading and editing LLC.
Round 2
Reviewer 2 Report
.